# BOOTSTRAPPED MODEL PREDICTIVE CONTROL

**Yuhang Wang, Hanwei Guo, Sizhe Wang, Long Qian, Xuguang Lan**[*]
National Key Laboratory of Human-Machine Hybrid Augmented Intelligence
National Engineering Research Center for Visual Information and Application
Institute of Artificial Intelligence and Robotics, Xi'an Jiaotong University, Xi'an, China
{wertyuilife,clannnagisa,wangsizhe,qianlongym}@stu.xjtu.edu.cn
xglan@mail.xjtu.edu.cn

## ABSTRACT

Model Predictive Control (MPC) has been demonstrated to be effective in continuous control tasks. When a world model and a value function are available, planning a sequence of actions ahead of time leads to a better policy. Existing methods typically obtain the value function and the corresponding policy in a model-free manner. However, we find that such an approach struggles with complex tasks, resulting in poor policy learning and inaccurate value estimation. To address this problem, we leverage the strengths of MPC itself. In this work, we introduce *Bootstrapped Model Predictive Control* (BMPC), a novel algorithm that performs policy learning in a bootstrapped manner. BMPC learns a network policy by imitating an MPC expert, and in turn, uses this policy to guide the MPC process. Combined with model-based TD-learning, our policy learning yields better value estimation and further boosts the efficiency of MPC. We also introduce a *lazy reanalyze* mechanism, which enables computationally efficient imitation learning. Our method achieves superior performance over prior works on diverse continuous control tasks. In particular, on challenging high-dimensional locomotion tasks, BMPC significantly improves data efficiency while also enhancing asymptotic performance and training stability, with comparable training time and smaller network sizes. Code is available at https://github.com/wertyuilife2/bmpc.

## 1 INTRODUCTION

Model-based reinforcement learning (MBRL) algorithms that incorporate online planning—often referred to as plan-based methods—have demonstrated superior performance and data efficiency across a range of domains, including chess (Silver et al., 2016; 2017; Schrittwieser et al., 2020), games (Ye et al., 2021), continuous control (Sikchi et al., 2022; Hansen et al., 2023), and the reasoning of large language models (Zhao et al., 2024; Putta et al., 2024). By leveraging future states and rewards predicted by a world model, planning algorithms can evaluate actions online and with greater accuracy, resulting in a better and more robust policy. This represents a key advantage of model-based planning, in contrast to model-free algorithms that learn a neural network directly through trial and error.

In the field of continuous control, model predictive control (MPC) has proven to be an effective planning approach (Lowrey et al., 2018; Hafner et al., 2019b; Hansen et al., 2022; Schubert et al., 2023). A notable example is TD-MPC2 (Hansen et al., 2023), an MPC-based MBRL algorithm with a robust world model, which demonstrates strong performance across a diverse range of continuous control tasks. Similar to existing methods (Bhardwaj et al., 2020; Sikchi et al., 2022), TD-MPC2 learns a network policy and a value function in a model-free manner. During inference, it uses policy-guided MPC for online planning, integrating the world model and the value function. However, our experiments reveal that despite having high-quality interaction data from MPC, the model-free policy learning struggles with challenging control tasks. The struggle in policy learning further indicates poor value learning, which can lead to inaccurate value estimation during MPC and degrade the overall performance of the MPC policy.

---

[*]Corresponding author.

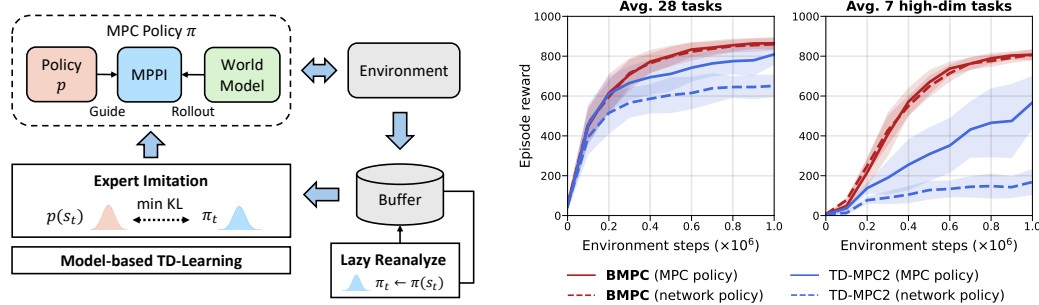

Figure 1: **Overview.** *(left)* BMPC learns a network policy through expert imitation and lazy reanalyze mechanism, planning during inference using guided MPC, and performs model-based value learning in an on-policy manner. *(right)* Averaged evaluation performance of the network policy compared to the MPC policy in BMPC and TD-MPC2 on DMControl tasks. BMPC achieves better policy learning, which further boosts the performance of MPC. Mean and 95% CIs over 5 seeds.

To address this problem, inspired by expert iteration algorithms (Anthony et al., 2017; Silver et al., 2017), we propose **B**ootstrapped **M**odel **P**redictive **C**ontrol (**BMPC**), which performs policy learning in a bootstrapped manner. We first execute MPC guided by the action sequences generated by a network policy, yielding a bootstrapped MPC expert. The network policy is then updated by imitating this expert, thus achieving policy improvement. By employing this iterative process, we leverage the capabilities of MPC planning to boost the efficiency of policy learning. For value learning, we compute TD-targets online using the world model to mitigate off-policy issues. For imitation target generation, we adopt a *lazy reanalyze* mechanism to maintain an expert action dataset, thereby supporting expert imitation from the MPC policy in a computationally efficient manner. An overview of BMPC is presented in Figure 1.

Through experiments, we show that learning a network policy through expert imitation can better leverage the strengths of MPC than learning a policy in a model-free manner, thus leading to better value estimation and MPC performance. Our method, BMPC, achieves superior sample efficiency over prior data-efficient RL methods across 42 continuous control tasks in DMControl (Tassa et al., 2018) and HumanoidBench (Sferrazza et al., 2024), with comparable training time and smaller network sizes. In particular, in challenging high-dimensional locomotion tasks, BMPC significantly improves data efficiency while also enhancing asymptotic performance and training stability. Additionally, our *lazy reanalyze* approach reduces the proportion of reanalyzed samples required by expert iteration algorithms (Ye et al., 2021; Wang et al., 2024) from 99% to 0.8%, while maintaining comparable policy learning efficiency. This avoids the need for extensive re-planning, significantly reducing the computational cost of BMPC.

## 2 RELATED WORK

**Model-based reinforcement learning.** MBRL focuses on using a model of the environment to help an agent make decisions, which typically involves learning a dynamic model and a reward model from data. To elaborate further, dyna-style approaches (Sutton, 1991; Janner et al., 2019; Hafner et al., 2019a; Okada & Taniguchi, 2022; Robine et al., 2023; Hafner et al., 2023a) use the model to simulate additional experiences based on real data, which improves the sample efficiency of the algorithm. In contrast, plan-based methods leverage the model for planning, resulting in better policies and further enhancing the sample efficiency of reinforcement learning. In the case of plan-based MBRL, if the model and value function are sufficiently accurate, planning alone can lead to a highly effective policy (Hafner et al., 2019b; Hansen et al., 2022; Schubert et al., 2023).

**Expert iteration.** Typically, planning algorithms need to roll out a large number of trajectories using the model to explore the solution space, which can be computationally intensive. Expert iteration methods (Anthony et al., 2017; Silver et al., 2017) improve the efficiency of planning by employing a network policy to guide the search direction. The planning algorithm, guided by the network policy, can be considered an expert, allowing the network policy to learn from it and thus achieve

policy improvement. By combining these two aspects, expert iteration can bootstrap the efficiency of planning and the capability of the network policy. Although tree-search-based expert iteration has achieved strong performance and data efficiency across a range of domains (Wang et al., 2024), in the domain of continuous control, pure MPC-based MBRL without expert iteration remains the superior approach (Hansen et al., 2023).

**Imitating and enhancing MPC.** Methods that aim to mimic or enhance MPC have gained significant attention in RL and robotics. Guided policy search methods (Levine & Koltun, 2013; Levine et al., 2016; Zhang et al., 2016; Sun et al., 2018) help RL policies explore effectively in challenging tasks by using an external plan-based policy, such as MPC, to guide the learning of a neural network policy. Pan et al. (2017; 2020); Sacks & Boots (2022); Fishman et al. (2023) propose methods where a neural network learns by mimicking an MPC controller, either to achieve a faster policy or to develop a more efficient neural planner. Alternatively, there are approaches that attempt to learn a residual policy on top of MPC (Silver et al., 2018; Sacks et al., 2024), which serves as another way of bootstrapping MPC. Additionally, Power & Berenson (2022); Sacks & Boots (2023) aim to improve MPC via learned sampling distributions. Unlike MBRL, the MPC policy in the aforementioned methods is derived from a real, carefully designed model, rather than a learned model. Moreover, these approaches do not adopt the iterative policy optimization scheme like expert iteration.

## 3 BACKGROUND

**Problem formulation.** We address reinforcement learning problems in continuous action spaces, which are modeled as an infinite-horizon Markov Decision Process (MDP). This MDP is defined by the tuple $(\mathcal{S}, \mathcal{A}, \mathcal{P}, \mathcal{R}, \gamma)$, where $\mathcal{S} \in \mathbb{R}^n$ and $\mathcal{A} \in \mathbb{R}^m$ are state and action spaces, $\mathbf{s} \in \mathcal{S}$ are states, $\mathbf{a} \in \mathcal{A}$ are actions, $\mathcal{P} : \mathcal{S} \times \mathcal{A} \mapsto \mathcal{S}$ is the state transition function, $\mathcal{R} : \mathcal{S} \times \mathcal{A} \mapsto \mathbb{R}$ is the reward function, and $\gamma$ is the discount factor. The objective in reinforcement learning is to derive a policy $\pi : \mathcal{S} \mapsto \mathcal{A}$ that maximizes the expected discounted cumulative reward, expressed as $\mathbb{E}_\pi \left[ \sum_{t=0}^{\infty} \gamma^t r_t \right]$, where $r_t = \mathcal{R}(\mathbf{s}_t, \pi(\mathbf{s}_t))$.

**TD-MPC2.** TD-MPC2 (Hansen et al., 2023) is a plan-based MBRL algorithm that learns a world model, a Q-function, and a corresponding policy, which are then used for MPC to derive an plan-based policy $\pi$. The model components of TD-MPC2 can be described by a tuple $(h, d, R, Q, p)$, where $\mathbf{z} = h(\mathbf{s})$ is the encoder that maps the observation $\mathbf{s}$ into a latent space vector $\mathbf{z}$, $\mathbf{z}' = d(\mathbf{z}, \mathbf{a})$ is the latent-space dynamics model, $\hat{r} = R(\mathbf{z}, \mathbf{a})$ is the reward prediction function, $\hat{q} = Q(\mathbf{z}, \mathbf{a})$ is the Q-value prediction function, and $\hat{a} = p(\mathbf{z})$ is the prior neural network policy. In this paper, we omit the representation of task embedding inputs of TD-MPC2, as we do not focus on its multi-task capabilities. Similar to model-free approaches like SAC (Haarnoja et al., 2018), TD-MPC2 learns the Q-function through iterations of the Bellman equation, and the network policy $p$ is optimized by maximizing the Q-value with entropy regularization, which can be formalized as the following update rules:

$$\phi \leftarrow \arg\min_{\phi} \mathbb{E}_{(\mathbf{s},\mathbf{a},r,\mathbf{s}')\sim\mathcal{B}} \left[ \mathrm{CE}(Q_\phi(\mathbf{z}, \mathbf{a}), r + \gamma Q_{\phi^-}(\mathbf{z}', p_\theta(\mathbf{z}'))) \right] \tag{1}$$

$$\theta \leftarrow \arg\max_{\theta} \mathbb{E}_{\mathbf{s}\sim\mathcal{B}} \left[ Q_\phi(\mathbf{z}, p_\theta(\mathbf{z})) + \beta \mathcal{H}(p_\theta(\cdot|\mathbf{z})) \right], \ \mathbf{z} = h(\mathbf{s}), \ \mathbf{z}' = h(\mathbf{s}') \tag{2}$$

where $\mathcal{H}$ is the entropy of $p$, $\beta$ is a hyperparameter for loss balancing, $\theta, \phi, \phi^-$ denote the parameters of the neural networks for $p$, $Q$, and the target Q-network, respectively. $\mathcal{B}$ represents the replay buffer. CE is the cross-entropy, used because TD-MPC2 formulates value prediction as a discrete regression problem. For simplicity, we omit the temporal expansion of the latent vector in update rules for TD-MPC2; for the full temporally weighted objectives, see Hansen et al. (2023).

**MPC with a policy prior.** During inference, TD-MPC2 performs MPC planning guided by the prior policy $p_\theta$. Specifically, TD-MPC2 leverages Model Predictive Path Integral (MPPI) (Williams et al., 2015) as its underlying MPC algorithm. MPPI models the action sequence $(\mathbf{a}_t, \mathbf{a}_{t+1}, ..., \mathbf{a}_{t+H})$ of length $H$ as being drawn from a time-dependent multivariate Gaussian with diagonal covariance, parameterized as $(\mu, \sigma)$, where $\mu, \sigma \in \mathbb{R}^{H \times m}$. During the planning process, MPPI iteratively samples sequences from $\mathcal{N}(\mu, \sigma^2)$, estimates their values by rolling out trajectories with the model, and updates $(\mu, \sigma)$ based on a weighted average of the top-k sequences, thus maximizing the expected estimated value of action sequence, which is expressed as:

$$\mu^*, \sigma^* = \arg\max_{\mu,\sigma} \mathbb{E}_{\mathbf{a}_{t:t+H}\sim\mathcal{N}(\mu,\sigma^2)} \left[ \hat{Q}(\mathbf{z}_t, \mathbf{a}_{t:t+H}) \right] \tag{3}$$

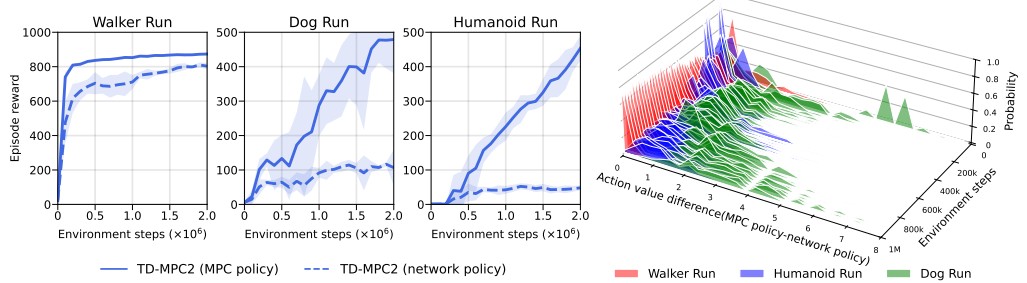

Figure 2: **Performance gap between TD-MPC2 policies.** *(left)* Evaluation performance of the network policy compared to the MPC policy in TD-MPC2. The network policy struggles with complex tasks like *Dog Run* and *Humanoid Run*. Mean and 95% CIs over 5 seeds. *(right)* Distributions of action value differences during MPPI over environment steps.

$$\hat{Q}(\mathbf{z}_t, \mathbf{a}_{t:t+H}) \doteq \gamma^H Q(\mathbf{z}_{t+H}, \mathbf{a}_{t+H}) + \sum_{h=0}^{H-1} \gamma^h R(\mathbf{z}_{t+h}, \mathbf{a}_{t+h}) \tag{4}$$

To integrate prior policy guidance, a portion of the sampled sequences is drawn from the distribution $p_\theta$. Ultimately, the first action of the optimized distribution $\mathbf{a}_t \sim \mathcal{N}(\mu_t^*, \sigma_t^*)$ is selected for execution during inference. For more details on the planning procedure, see Hansen et al. (2022).

In this paper, we adopt the same world model architecture and MPPI algorithm as TD-MPC2, and we use similar mathematical notations to describe our algorithm components.

# 4 METHOD

## 4.1 INSUFFICIENT POLICY LEARNING IN MODEL-FREE APPROACH

A model-free approach is adopted in TD-MPC2 to learn both the Q-function and a max-Q network policy. During inference, it leverages MPC to plan actions, guided by the network policy. Since the MPC policy—based on the learned model and online planning—typically outperforms the network policy, it provides higher-quality samples for model-free learning, thereby improving the efficiency of both policy and value learning. However, we find that in challenging environments, even with high-quality samples from the MPC policy, the model-free approach still struggles, leading to a performance gap between the MPC policy and the network policy. This gap indicates inaccurate value estimation, further degrading the planning performance of MPC.

Figure 2 shows the evaluation performance of both the network policy and MPC policy during training, on three locomotion tasks in DMControl (Tassa et al., 2018): *Walker Run*, *Humanoid Run*, and *Dog Run*. In all three tasks, a policy performance gap is evident, though the extent of the gap varies. In simpler task *Walker Run*, the network policy performs comparably to the MPC policy. However, in more complex tasks like *Humanoid Run* and *Dog Run*, the network policy struggles to improve, while the MPC policy maintains high performance.

To better understand the root cause of this performance gap, we analyze the planning process of MPPI. We represent the prior action sequences generated by the network policy as $\{\mathbf{a}_{t:t+H}^{i,p_\theta}\}_{i=1}^n$, where $n$ is the number of prior action sequences, and the final action sequence selected by MPPI as $\mathbf{a}_{t:t+H}^{\mathrm{mpc}}$. To quantify the optimization achieved by MPPI, we compute the difference between the value of the final selected sequence and the average value of the prior sequences:

$$\Delta\hat{Q}(\mathbf{z}_t) \doteq \hat{Q}(\mathbf{z}_t, \mathbf{a}_{t:t+H}^{\mathrm{mpc}}) - \frac{1}{n}\sum_{i=1}^n \hat{Q}(\mathbf{z}_t, \mathbf{a}_{t:t+H}^{i,p_\theta}) \tag{5}$$

where $\hat{Q}$ denotes the value estimated by the model in Equation 4. Overall, $\Delta\hat{Q}(\mathbf{z}_t)$ reflects the improvements that MPC makes to action value at time-step $t$. As illustrated in Figure 2, we plot the distributions of $\Delta\hat{Q}$ over training steps within an episode. The figure shows that improvements

brought by MPC are continuously increasing in *Humanoid Run* and *Dog Run*, whereas in *Walker Run*, the network policy already performs well, leaving little room for MPC to further optimize.

These experiments demonstrate that, although TD-MPC2 exhibits strong performance in high-dimensional locomotion tasks, much of its performance is due to the use of online planning. In contrast, network policy learning struggles considerably with these tasks. Consequently, we argue that this inefficient policy learning indicates poor value learning, reducing the data efficiency of the algorithm. Specifically, because the value network is learned to represent the value of the corresponding network policy, this gap implies that the terminal value used by MPC is derived from a *weaker policy*, which will result in inaccurate value estimation during planning. A natural solution, which we propose, is to leverage the imitation of the MPC expert to achieve policy learning.

## 4.2 BOOTSTRAPPED MODEL PREDICTIVE CONTROL

We propose **BMPC**, a plan-based MBRL algorithm based on TD-MPC2's world model. BMPC learns a neural network policy by imitating an MPC expert, and in turn, uses this policy to guide the MPC process. The world model is leveraged in BMPC to perform on-policy TD-learning of a value network, which is used for terminal value calculation during MPC. Additionally, we introduce a *lazy reanalyze* mechanism to maintain an expert dataset for more computationally efficient imitation learning. The algorithm for BMPC training is presented in Algorithm 1.

**Policy learning through expert imitation.** BMPC uses a prior policy $p_\theta$ to guide the planning process of MPC. In other words, we can describe this as MPC bootstrapping $p_\theta$ into an expert policy, denoted as $\pi(\cdot|\mathbf{z}, p_\theta)$. Thus, we learn the policy $p_\theta$ by imitating $\pi$, which can be formalized as the following objective:

$$\mathcal{L}_p(\theta) \doteq \mathbb{E}_{(\mathbf{s},\mathbf{a})_{0:H} \sim \mathcal{B}} \left[ \sum_{t=0}^{H} \lambda^t \left[ \mathrm{KL}(\pi(\cdot|h(\mathbf{s}_t), p_\theta), p_\theta(\cdot|\mathbf{z}_t))/\max(1, S) - \beta \mathcal{H}(p_\theta(\cdot|\mathbf{z}_t)) \right] \right] \quad (6)$$
$$\mathbf{z}_0 = h(\mathbf{s}_0), \ \mathbf{z}_{t+1} = d(\mathbf{z}_t, \mathbf{a}_t), \ S \doteq \mathrm{EMA}(\mathrm{Per}(\mathrm{KL}(\pi, p_\theta), 95) - \mathrm{Per}(\mathrm{KL}(\pi, p_\theta), 5), 0.99)$$

where $\mathcal{H}$ is the entropy, KL is the Kullback–Leibler divergence, $\mathbf{z}_{0:H}$ are latent vectors rolled out through model $h$ and $d$. $\beta$ and $\lambda$ are hyperparameters for loss balancing and temporal weighting, respectively. Empirically, when action space is large, imitating the action distribution $\pi$ is more efficient than imitating the exact actions $\mathbf{a} \sim \pi$, especially when both the student and expert policies' distributions belong to the same parametric family. Since the MPC policy is parameterized as a multivariate Gaussian, we choose to parameterize the neural network policy as a multivariate Gaussian as well, allowing us to compute the KL divergence in closed form. As the KL divergence between multivariate Gaussian distributions can vary significantly across tasks and action spaces, affecting training stability, we normalize the KL loss using moving percentiles $S$ to keep the loss value within an acceptable range. This method is also commonly used to balance the policy loss and entropy loss (Hafner et al., 2023a; Hansen et al., 2023).

**Model-based TD-learning.** Since we do not employ a SAC-style max-Q approach for policy improvement, we opt to learn a state value function $V_\phi$ instead of a state-action value function $Q_\phi$. We construct an n-step TD-target $\hat{V}$ using the latest model, policy, and target value network. The value network learns to minimize the cross-entropy loss with respect to the discretized TD-target:

$$\mathcal{L}_V(\phi) \doteq \mathbb{E}_{(\mathbf{s},\mathbf{a})_{0:H} \sim \mathcal{B}} \left[ \sum_{t=0}^{H} \lambda^t \left[ \mathrm{CE}(V_\phi(\mathbf{z}_t), \hat{V}(h(\mathbf{s}_t))) \right] \right], \ \mathbf{z}_0 = h(\mathbf{s}_0), \ \mathbf{z}_{t+1} = d(\mathbf{z}_t, \mathbf{a}_t)$$
$$\hat{V}(\mathbf{z}'_t) \doteq \gamma^N V_{\phi^-}(\mathbf{z}'_{t+N}) + \sum_{k=0}^{N-1} \gamma^k R(\mathbf{z}'_{t+k}, p_\theta(\mathbf{z}'_{t+k})), \ \mathbf{z}'_{t+1} = d(\mathbf{z}'_t, p_\theta(\mathbf{z}'_t)) \quad (7)$$

where $N$ is the TD horizon, $\mathbf{z}_{0:H}$ are latent vectors rolled out through model $h$ and $d$. $\hat{V}$ is the TD-target computed using the model $d$, $R$ and the policy $p_\theta$ in an on-policy manner. In practice, we found that $N = 1$ is a more suitable choice, likely because the world model of TD-MPC2 is trained with a short horizon ($H = 3, \lambda = 0.5$), limiting its ability to predict rewards over long sequences. As a result, setting $N$ too large would lead to excessive compounding errors.

Indeed, using the original Q-iteration method for value learning is also a feasible choice. However, due to the changes in the policy learning approach, this option introduces certain off-policy issues, which can lead to lower data efficiency and unstable training in tasks where the policy varies significantly during training. We compare the results of the two value learning approaches in Section 5.1.

**Lazy reanalyze.** In practice, it is costly to compute the policy objective 6 directly, as it requires re-planning for all samples during every update, which is infeasible for MPC algorithms. Instead, we choose to maintain imitation targets in the replay buffer through *lazy reanalyze*, thus resulting in a surrogate policy objective 8.

During every $k$-th network update, we draw $b$ samples from the batch, re-plan them, and obtain fresh imitation targets, i.e., the mean and standard deviation of the action distribution $\pi_t = \pi(\cdot|h(\mathbf{s}_t), p_\theta)$. These targets $\pi_t$ are then placed back into the replay buffer. This reanalyzing process is performed independently of the training process. Thus, we can approximately regard the replay buffer as an expert dataset, and directly sample state-action pairs from it for supervised learning. To increase exploration in MPC planning, we additionally add noise to the prior policy during re-planning.

This approach is inspired by the *reanalyze* proposed in Schrittwieser et al. (2020), a common method in sample-efficient expert iteration algorithms (Ye et al., 2021; Wang et al., 2024). The key difference is that *reanalyze* performs re-planning during every network update, with 99%-100% of samples re-planned (i.e., reanalyze ratio). These reanalyzed samples are immediately used for imitation learning and then discarded. In contrast, *lazy reanalyze* performs far fewer re-plannings and places the reanalyzed samples back into the buffer for reuse.

---

**Algorithm 1** BMPC training

**Require:** Initialize $p_\theta, V_\phi, h, d, R$ randomly.
    $\mathcal{B}, k$: replay buffer, lazy reanalyze interval.
1: **while** not converged **do**
2:     *// Collect experience*
3:     **for** step $t = 0...T$ **do**
4:         $\pi_t \leftarrow \pi(\cdot|h(\mathbf{s}_t), p_\theta)$
5:         $\mathbf{a}_t \sim \pi_t$
6:         $(\mathbf{s}_{t+1}, r_t) \leftarrow \text{env.step}(\mathbf{a}_t)$
7:         $\mathcal{B} \leftarrow \mathcal{B} \cup (\mathbf{s}_t, \mathbf{a}_t, r_t, \mathbf{s}_{t+1}, \pi_t)$
8:     *// Update networks*
9:     **for** num updates per episode **do**
10:         $\{\mathbf{s}_t, \mathbf{a}_t, r_t, \mathbf{s}_{t+1}, \pi_t\}_{t:t+H} \sim \mathcal{B}$
11:         Update $h, d, R$ as in TD-MPC2.
12:         Update $V_\phi$ via Equation 7.
13:         Update $p_\theta$ via Equation 8.
14:         Update $V_{\phi^-}$ via EMA.
15:     *// Lazy reanalyze*
16:         **if** update_step $\% k == 0$ **then**
17:             $\pi_{t:t+H} \leftarrow \pi(\cdot|h(\mathbf{s}_{t:t+H}), p_\theta)$
18:         $\mathcal{B} \xleftarrow{\text{update}} \pi_{t:t+H}$

---

Specifically, the reanalyze interval $k$ and the reanalyze batch size $b$ are hyperparameters, where we choose $k = 10$ and $b = 20$, with a batch size of 256 for network updates. Under this setup, *lazy reanalyze* achieves a computational cost equivalent to a reanalyze ratio of 0.8%, which is over 100 times lower than the typical reanalyze ratio of 99% (Ye et al., 2021; Wang et al., 2024). When combined with batched MPPI planning on GPU, *lazy reanalyze* introduces only a 10%-20% increase in training wall-time.

The surrogate policy objective with *lazy reanalyze* can be formalized as:

$$\mathcal{L}_p^{lazy}(\theta) \doteq \mathop{\mathbb{E}}_{(\mathbf{s},\mathbf{a},\pi)_{0:H} \sim B} \left[ \sum_{t=0}^{H} \lambda^t \left[ \text{KL}(\pi_t, p_\theta(\cdot|\mathbf{z}_t))/\max(1, S) - \beta\mathcal{H}(p_\theta(\cdot|\mathbf{z}_t)) \right] \right] \qquad (8)$$

where $\pi_t$ is the expert action distribution we maintain in the replay buffer.

## 5 EXPERIMENTS

In this paper, we propose BMPC to more effectively leverage the strengths of MPC in continuous control tasks. Our approach integrates expert imitation for policy learning, performs model-based TD-learning for value learning, and introduces *lazy reanalyze* to better utilize re-planning results. Through our experiments, we aim to answer the following key questions:

- How does BMPC perform as a data-efficient continuous control algorithm compared to the current state-of-the-art methods?
- Does BMPC lead to better policy learning, and how can this be further leveraged?

- How does *lazy reanalyze* affect the performance and training time of BMPC?

To ensure a direct comparison, we evaluate BMPC on 28 DMControl (Tassa et al., 2018) tasks used in the TD-MPC2 (Hansen et al., 2023) [1], and 14 tasks from HumanoidBench (Sferrazza et al., 2024) locomotion suite. The tasks covers a diverse range of continuous control challenges, including sparse reward, locomotion with high-dimensional state and action space (up to $\mathcal{A} \in \mathbb{R}^{61}$). Visualizations of the tasks can be found in Appendix D. To avoid ambiguity, we use the term "environment step" in our experiments, where an environment step refers to the number of inference steps multiplied by the action repeat (2 in DMControl, 1 in HumanoidBench).

**Baselines.** We select three state-of-the-art data-efficient RL methods as baselines: (1) SAC (Haarnoja et al., 2018), a model-free actor-critic algorithm rooted in maximum entropy reinforcement learning; (2) DreamerV3 (Hafner et al., 2023a), a Dyna-style MBRL algorithm that trains a model through reconstruction loss and learns a model-free policy from trajectories imagined by the model; (3) TD-MPC2 (Hansen et al., 2023), an MPC-based MBRL algorithm that learns a network policy and Q-function in a model-free manner, and performs MPC planning at inference based on them and the model. Both MBRL methods learn an implicit model to roll out sequences in latent space and predict rewards.

For SAC, we use the results from TD-MPC2 and HumanoidBench; For DreamerV3, we use its default settings on DMControl tasks, corresponding to a network size of 12M, and use the results from HumanoidBench; For TD-MPC2, we use its default configuration, corresponding to a network with 5M parameters. For BMPC, we adopt a network almost identical to TD-MPC2, except that we replace the Q-network with a V-network. Notably, as we do not use a Q-based method, we find that BMPC's performance is less dependent on ensemble networks, allowing us to reduce the default 5 ensemble value networks to 2. This results in a smaller network for BMPC, with only 3M parameters. We use the same hyperparameters for BMPC across all tasks, see Table 2, and detailed baseline configurations are provided in Appendix B.

## 5.1 RESULTS

**Benchmark performance.** We first compare BMPC against the baselines across all 28 DMControl tasks. Due to limited space, we only present results for 10 selected tasks, as shown in Figure 3. The training curves for all tasks are provided in Appendix C. Our results show that BMPC consistently achieves either superior or comparable performance relative to the baselines on most tasks. Notably, on high-dimensional locomotion tasks, such as *Dog* and *Humanoid*, BMPC shows significant improvements in data efficiency, despite having fewer learnable parameters. The results on DMControl indicate that BMPC maintains the performance of TD-MPC2 across a wide range of control tasks, while significantly enhancing performance in tasks where model-free approach struggles.

We further compare BMPC with the baselines on the 7 high-dimensional tasks, as shown in Figure 4. The environment steps are extended from 1M to 4M for a comprehensive comparison. In addition to its improved data efficiency at short training lengths, BMPC also outperforms the baselines in terms of asymptotic performance and training stability, as indicated by the confidence intervals of the curves. Finally, the average steps-to-solve (the number of steps required to achieve 795, which is 90% of the asymptotic reward) across the 7 tasks for BMPC is 90k steps, while for TD-MPC2 it is 360k steps, indicating a 300% increase in data efficiency.

On the HumanoidBench locomotion suite, which requires the agent to control a more complex embodiment—a Unitree robot with a large action space ($\mathcal{A} \in \mathbb{R}^{61}$)—BMPC maintains superior performance compared to baselines, further demonstrating its advantage on challenging high-dimensional tasks. We present results for all tasks in Figure 5.

**Leveraging better policy learning.** In BMPC, we obtain a network policy through expert imitation. To verify whether this network policy is better and how we can leverage it, we design ablation experiments as shown in Figure 6 and Figure 7a.

Figure 6 illustrates the performance gap between the network policy and MPC policy for BMPC and TD-MPC2 on DMControl, similar to Figure 2. Training curves for all tasks are provided in Appendix C. Surprisingly, through expert imitation, BMPC enables its network policy to perform *nearly on*

---

[1]Excluding custom tasks created for multitask training.

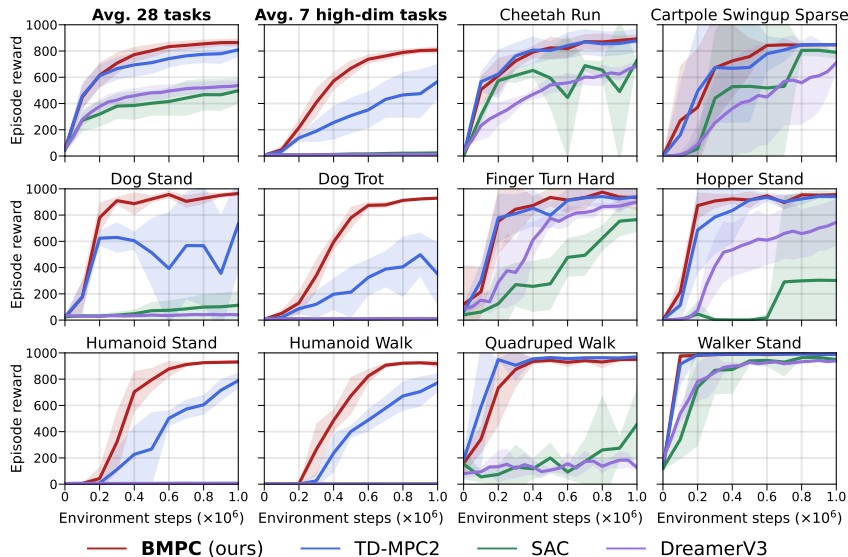

Figure 3: **DMControl tasks.** Comparing BMPC to baselines on DMControl tasks. In the top left, we present the average performance of 7 high-dimensional locomotion tasks and all 28 tasks. Mean and 95% CIs over 5 seeds[2]. Training curves for all tasks are provided in Appendix C.

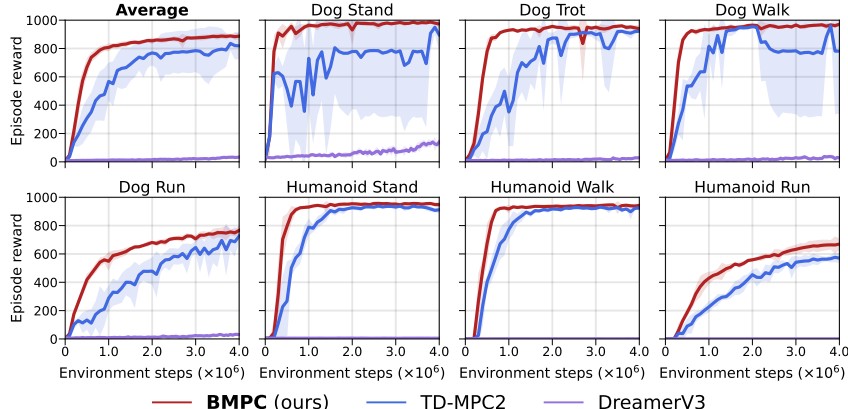

Figure 4: **High-dimensional locomotion tasks.** Comparison of BMPC with baselines on the 7 most challenging high-dimensional locomotion tasks; The environment steps are extended to 4M for a comprehensive comparison. In the top left, we present the results averaged over all 7 tasks. Mean and 95% CIs over 5 seeds.

*par* with its MPC policy. In contrast, TD-MPC2 exhibits a substantial performance gap between its network policy and MPC policy, particularly in challenging tasks such as *Dog* and *Humanoid*. This indicates that BMPC's network policy can serve as a *viable final inference strategy* without the need for online planning, which is suitable for real-time control tasks that demand low latency.

We further conduct ablation studies to show how this improved network policy contributes to BMPC's performance. We introduce three BMPC variants for comparison: (1) *Variant 1*: based on TD-MPC2, we use expert imitation to *additionally* learn a network policy, which is used to guide MPPI planning, while value learning still relies on the original policy; (2) *Variant 2*: based on *Variant 1*, we use both two network policies to guide MPPI planning simultaneously; (3) *Variant 3*: we *replace* TD-MPC2's policy learning approach with expert imitation, thus changing the policy used in value learning, while still learning value based on Q-iteration; (4) BMPC: based on *Variant 3*, we adopt model-based on-policy TD-learning. For further details of these variants, see Appendix A.

---

[2]Except SAC, which uses 3 seeds.

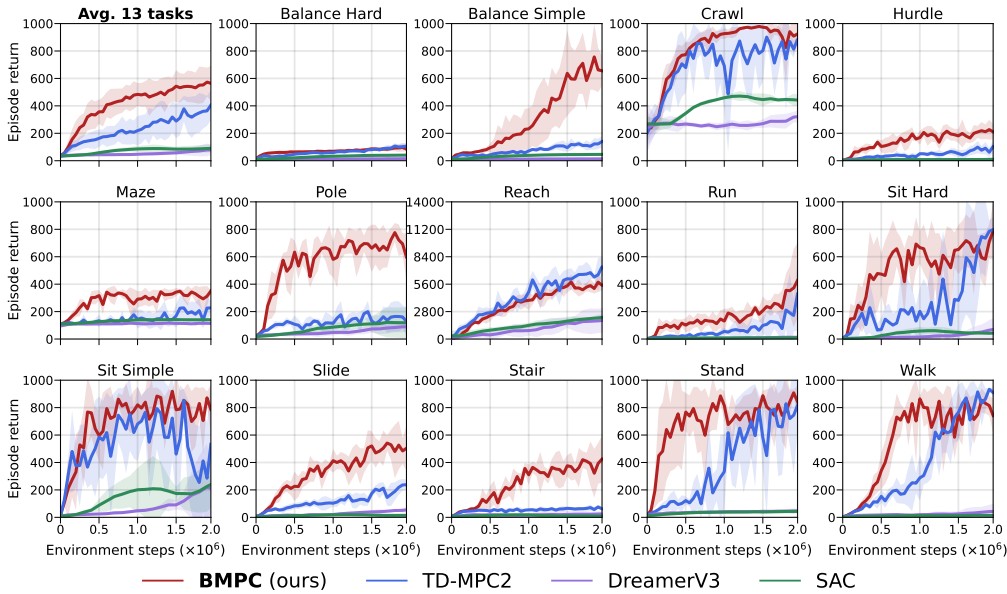

Figure 5: **HumanoidBench locomotion suite.** Comparing BMPC to baselines on HumanoidBench locomotion suite. In the top left, we present the average performance of all 13 tasks except for Reach due to the different reward scales. Mean and 95% CIs over 5 seeds.

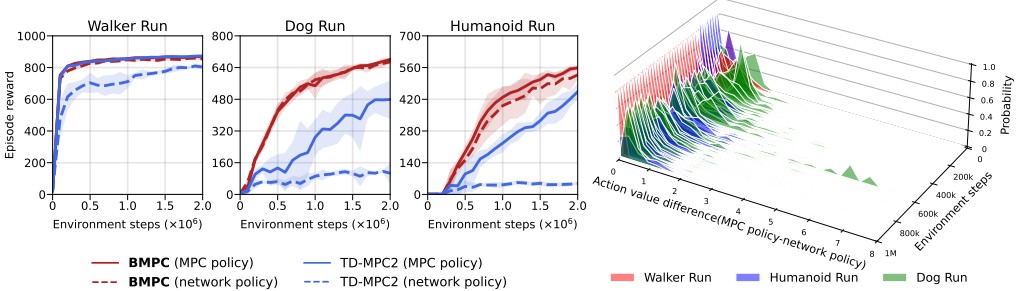

Figure 6: **Performance gap between BMPC policies.** *(left)* Evaluation performance of the network policy compared to the MPC policy in BMPC and TD-MPC2, The performance gap of BMPC is barely noticeable. Mean and 95% CIs over 5 seeds. Curves for all tasks are provided in Appendix C. *(right)* Distributions of action value differences during MPPI over environment steps for BMPC.

Figure 7a shows the results of BMPC variants on the *Dog Run* and *Humanoid Run*. We find that even with a better policy, using it to guide MPC does not result in improved performance, as shown by the results of *Variant 1* and *Variant 2*. The key to performance improvement lies in using this policy to learn a better value function, as indicated by the results of *Variant 3* and BMPC. Additionally, model-based value learning avoids off-policy issues, leading to improved data efficiency.

**Lazy reanalyze ablation.** BMPC maintains an expert dataset through *lazy reanalyze* to achieve computationally efficient expert imitation. We explore the relationship between the frequency of re-planning and performance by evaluating lazy reanalyze interval $k \in \{10, 40, 80, \infty\}$, corresponding to reanalyze ratios of 0.8%, 0.2%, 0.1%, and 0%, respectively. The results, shown in Figure 7b, indicate that the reanalyze interval $k$ considerably affects BMPC's performance. However, as $k$ decreases, the impact becomes less pronounced. Beyond $k = 10$, further increasing the reanalyze frequency does not significantly improve performance but results in substantial computational overhead. Thus, we set $k = 10$ as the default value. It is worth noting that in all our experiments, the replay buffer size is 1M, which is quite large for an expert dataset, but even with low-frequency reanalyzing, the freshness of the data is sufficient to support expert imitation.

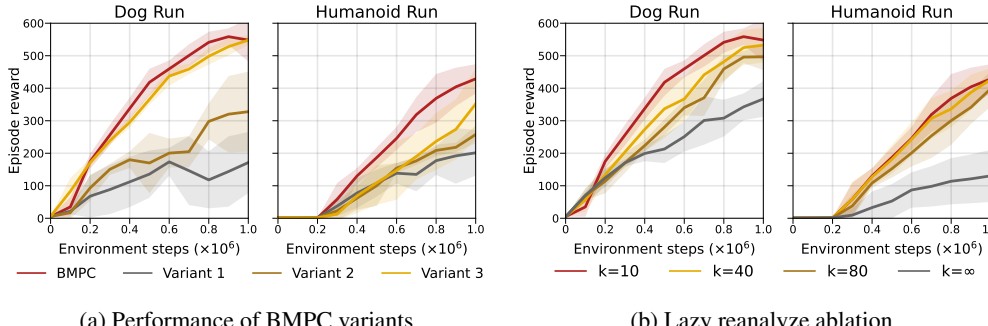

(a) Performance of BMPC variants      (b) Lazy reanalyze ablation

Figure 7: **Ablations.** (a) Performance of BMPC variants with different ways to leverage the network policy, demonstrating the importance of value learning. (b) Performance of BMPC with different lazy reanalyze interval $k$; Mean and 95% CIs over 5 seeds.

**Training wall-time.** We compare the training wall-time and time-to-solve of BMPC and TD-MPC2 in *Walker Walk* and *Dog Walk*, where time-to-solve is defined as the time required to achieve rewards of 899 and 872, respectively (90% of the asymptotic reward), as shown in Table 1. The experiments are conducted using a single RTX 3090 GPU. BMPC and TD-MPC2 have similar training times per 500k steps, although *lazy reanalyze* increases training time by approximately 20%, this increase is offset by the reduced size of the network. Due to its high data efficiency, the time-to-solve of BMPC is 2 times shorter than TD-MPC2 on *Dog Walk*.

Notably, our BMPC implementation uses only a single thread for training and does not fully parallelize the *lazy reanalyze* process. Since *lazy reanalyze* operates independently of network training, it could be parallelized using a separate thread to further reduce training time. Additionally, BMPC's network policy performs nearly on par with MPC planning on DMControl tasks, making it feasible to use the network policy for inference without planning, which would significantly reduce inference time.

Table 1: **Wall-time.** Time-to-solve and time per 500k environment steps for the *Walker Walk* and *Dog Walk*. Mean of 3 runs.

| *Wall-time*(h) | Walker Walk | | Dog Walk | |
|---|---|---|---|---|
| | TD-MPC2 | **BMPC** | TD-MPC2 | **BMPC** |
| time-to-solve | 0.40 | 0.43 | 2.03 | 0.87 |
| h/500k steps | 7.67 | 7.32 | 8.47 | 8.71 |

## 6   CONCLUSIONS AND FUTURE DIRECTIONS

In this paper, we introduce BMPC, a plan-based MBRL algorithm that leverages the strong performance of MPC to achieve better policy and value learning in continuous control tasks. Our approach demonstrates superior performance, particularly in complex locomotion tasks, while maintaining a comparable training time and a smaller network size compared to state-of-the-art methods.

Due to the current world model's lack of long-horizon prediction capability, BMPC is limited to a TD-horizon of 1. As the model capability improves, there is potential to extend the TD-horizon for enhanced value learning. Moreover, BMPC can further benefit from advanced expert iteration techniques, such as the plan-based value estimation proposed in Wang et al. (2024).

It is also worth exploring the combination of expert imitation with the max-Q gradient for joint policy improvement, integrating the strengths of both approaches, such as by combining loss functions or integrating independent value functions for planning.

Finally, BMPC can also be applied in multi-task and offline settings. Although experiments in these areas have yet to be conducted, we believe BMPC can further harness the capabilities of MPC.

## 7   REPRODUCIBILITY STATEMENT

We have open-sourced our work, available at https://github.com/wertyuilife2/bmpc. By running the code with the default configuration, the results presented in this paper can be reproduced.

ACKNOWLEDGMENTS

This work was supported in part by NSFC under grant No.62125305, No.U23A20339, No.62088102, No.62203348.

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

## A    IMPLEMENTATION DETAILS

**Exploration in lazy reanalyze.** The network policy of BMPC uses a neural network to obtain the mean and log standard deviation of a Gaussian distribution. The mean is derived by applying the tanh function to squash the output. The log standard deviation is first squashed to the range $[-1, 1]$ using the tanh function, and then mapped linearly to the range $[\mathsf{log\_std\_min}, \mathsf{log\_std\_max}]$.

During *lazy reanalyze*, we increase the value of $\mathsf{log\_std\_min}$ to enhance exploration in MPC. Specifically, the default values are $\mathsf{log\_std\_min} = -3$ and $\mathsf{log\_std\_max} = 1$. When reanalyzing, we set $\mathsf{log\_std\_min} = -2$ and $\mathsf{log\_std\_max} = 1$. Mathematically, this adjustment is equivalent to $\mathsf{log\_std}_{reanalyze} = \mathsf{log\_std} \times 0.75 + 0.25$.

This mechanism does not increase overall performance but helps prevent BMPC from prematurely converging to local optima due to iterative policy learning. A larger policy std during *lazy reanalyze* may further improve the exploration of MPC, but this would likely require increasing the number of MPPI iterations to ensure convergence, which we opt not to do.

**Policy loss.** In our experiments, we try both log probability loss in Wang et al. (2024) and KL loss for expert imitation, observing substantial differences in the policy std. We find that network policy accurately mimics the std of the MPC policy through KL loss; while using log probability loss tends to result in a large policy std, affecting the optimality of the algorithm. We speculate this may be due to the use of *lazy reanalyze*, leading the expert action dataset to encompass behaviors of different MPC policies over a longer period, necessitating a larger std for the network policy. Alternatively, it may result from the MPC policy's tendency to produce "shaky actions" [3], leading to a larger policy std when using log probability loss.

**Hyperparameters.** We use the same hyperparameters for all tasks. BMPC is based on the world model and MPPI of TD-MPC2. For the hyperparameters in these components, we use the same default values as those in TD-MPC2 (Hansen et al., 2023). The hyperparameters of BMPC are detailed in Table 2.

---

[3]see https://github.com/nicklashansen/tdmpc2/issues/26

Table 2: **BMPC Hyperparameters.** We use the same hyperparameters for all tasks.

| Hyperparameter | Value |
|---|---|
| Entropy loss coef. | $1 \times 10^{-4}$ |
| Batch size | 256 |
| TD horizon ($N$) | 1 |
| Number of ensemble value networks | 2 |
| Lazy reanalyze interval ($k$) | 10 |
| Lazy reanalyze batch size ($b$) | 20 |
| Planning horizon | 3 |
| Re-planning horizon | 3 |
| Policy log std. min. | -3 |
| Policy log std. max. | 1 |
| Policy log std. min. (re-planning) | -2 |
| Policy log std. max. (re-planning) | 1 |

**BMPC variants.** We conduct ablation studies by introducing three BMPC variants in the Experiments section. Below, we provide a detailed explanation of these variants along with their respective design choices.

*Key Facts:* The design of BMPC and its variants is guided by the following components:

- A network policy can be learned using:
    - (1a) Max-Q gradient-based learning;
    - (1b) Expert imitation.
- A network policy is used for:
    - (2a) Computing the TD-target during value learning;
    - (2b) Guiding the planning process of MPPI.
- The value function can be learned using:
    - (3a) Off-policy TD-learning (Q-iteration);
    - (3b) On-policy model-based TD-learning.

*BMPC Variants:* We define the following three BMPC variants based on different combinations of the above components:

- *Variant 1*:
    - Learn network policy $A$ using (1a) and network policy $B$ using (1b).
    - Use policy $A$ for (2a) and policy $B$ for (2b).
    - Learn the value function using (3a).
- *Variant 2*:
    - Learn network policy $A$ using (1a) and network policy $B$ using (1b).
    - Use policy $A$ for (2a) and both policies $A$ and $B$ for (2b).
    - Learn the value function using (3a).
- *Variant 3*:
    - Learn network policy $A$ using (1b).
    - Use policy $A$ for both (2a) and (2b).
    - Learn the value function using (3a).

*BMPC:* For comparison, our proposed BMPC approach is defined as follows:

- Learn network policy $A$ using (1b).
- Use policy $A$ for both (2a) and (2b).
- Learn the value function using (3b).

*Additional Remarks:* For all variants, we use 5 ensemble Q-networks instead of 2 as in BMPC. For *Variant 2*, we generate 24 guiding trajectories from each policy ($A$ and $B$), and increase the number of elite trajectories in the MPPI process from 64 to 88.

## B  BASELINES DETAILS

We report our implementation details of baselines in this section.

**SAC.** We use the results from TD-MPC2 [4] and HumanoidBench [5] for DMControl and Humanoid-Bench, respectively. For the hyperparameters of SAC, see Hansen et al. (2023) and Sferrazza et al. (2024).

**DreamerV3.** For DMControl, we use the latest implementation[6], referencing Hafner et al. (2023a), which differs from the older version Hafner et al. (2023b). We use the default settings on DMC proprio tasks, corresponding to a network size of 12M and a UTD ratio of 512. We discover that the performance of DreamerV3 diverges from what is reported in Hansen et al. (2023). For example, the performance on the *Fish Swim* has improved, while the performance on the *Walker Run* has decreased. This is likely because the newer version of DreamerV3 changes the policy learning approach in continuous action space, from stochastic backpropagation to the Reinforce. However, since both results underperform relative to TD-MPC2, this does not affect the overall comparison. For a comprehensive list of hyperparameters, please refer to the original paper (Hafner et al., 2023a). For HumanoidBench, we use the results from HumanoidBench repository[5].

**TD-MPC2.** For both DMControl and HumanoidBench, we use the latest code with its default hyperparameters[4]. For a comprehensive list of hyperparameters, please refer to their original paper(Hansen et al., 2023).

## C  ALL TRAINING CURVES

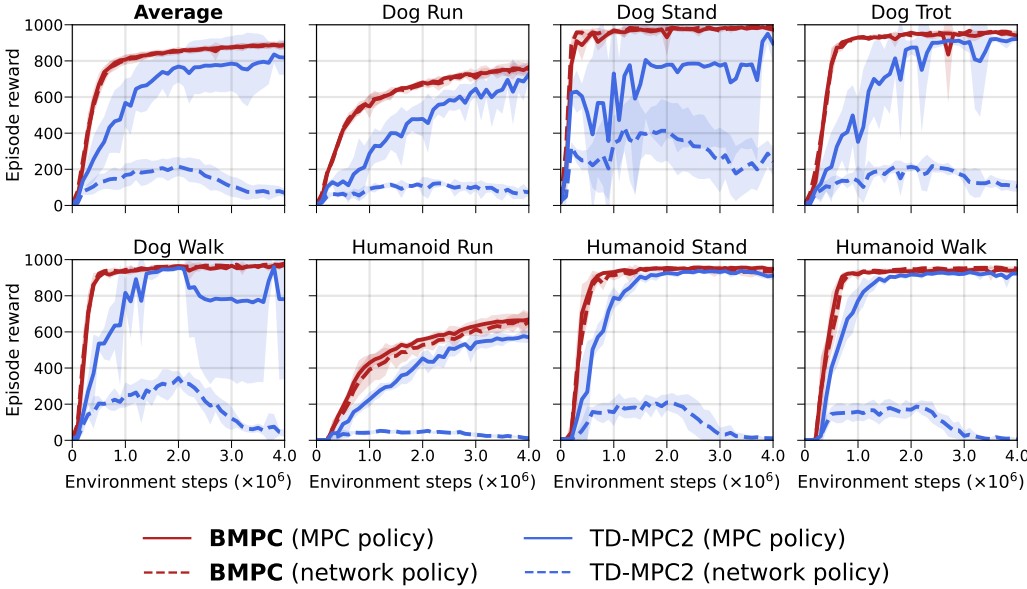

Figure 8: **Performance of different policies on high-dimensional locomotion tasks.** Evaluation performance of the network policy compared to the MPC policy in BMPC and TD-MPC2 on the 7 high-dimensional locomotion tasks. The environment steps are extended to 4M for a comprehensive comparison. In the top left, we present the average performance. Mean and 95% CIs over 5 seeds.

---

[4]We use results and code in https://github.com/nicklashansen/tdmpc2.

[5]We use results in https://github.com/carlosferrazza/humanoid-bench.

[6]We use the code in https://github.com/danijar/dreamerv3.

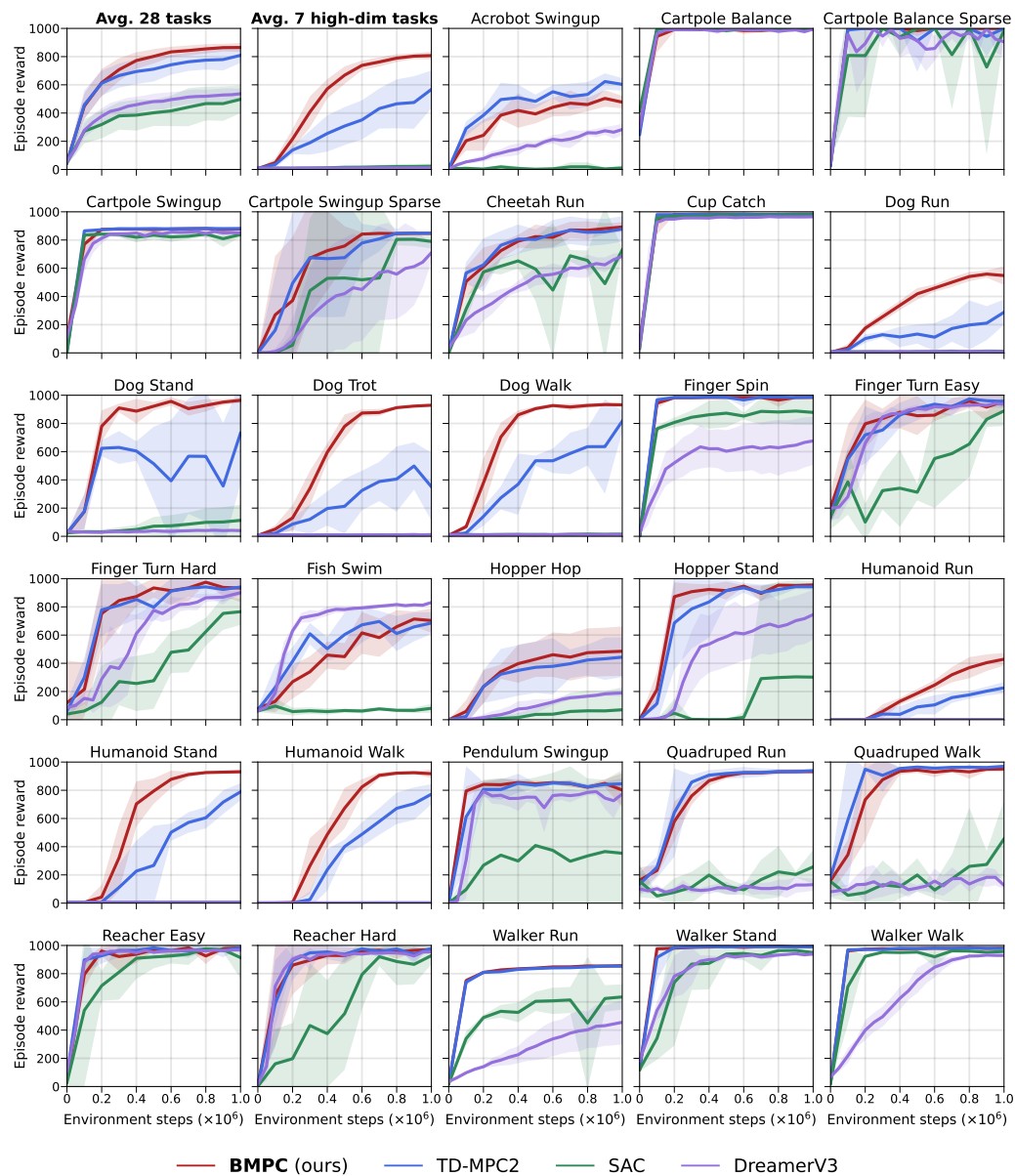

Figure 9: **All DMControl tasks.** Comparing BMPC to baselines on DMControl tasks. In the top left, we present the average performance of 7 high-dimensional locomotion tasks and all 28 tasks. Mean and 95% CIs over 5 seeds[7].

---

[7]Except SAC, which uses 3 seeds.

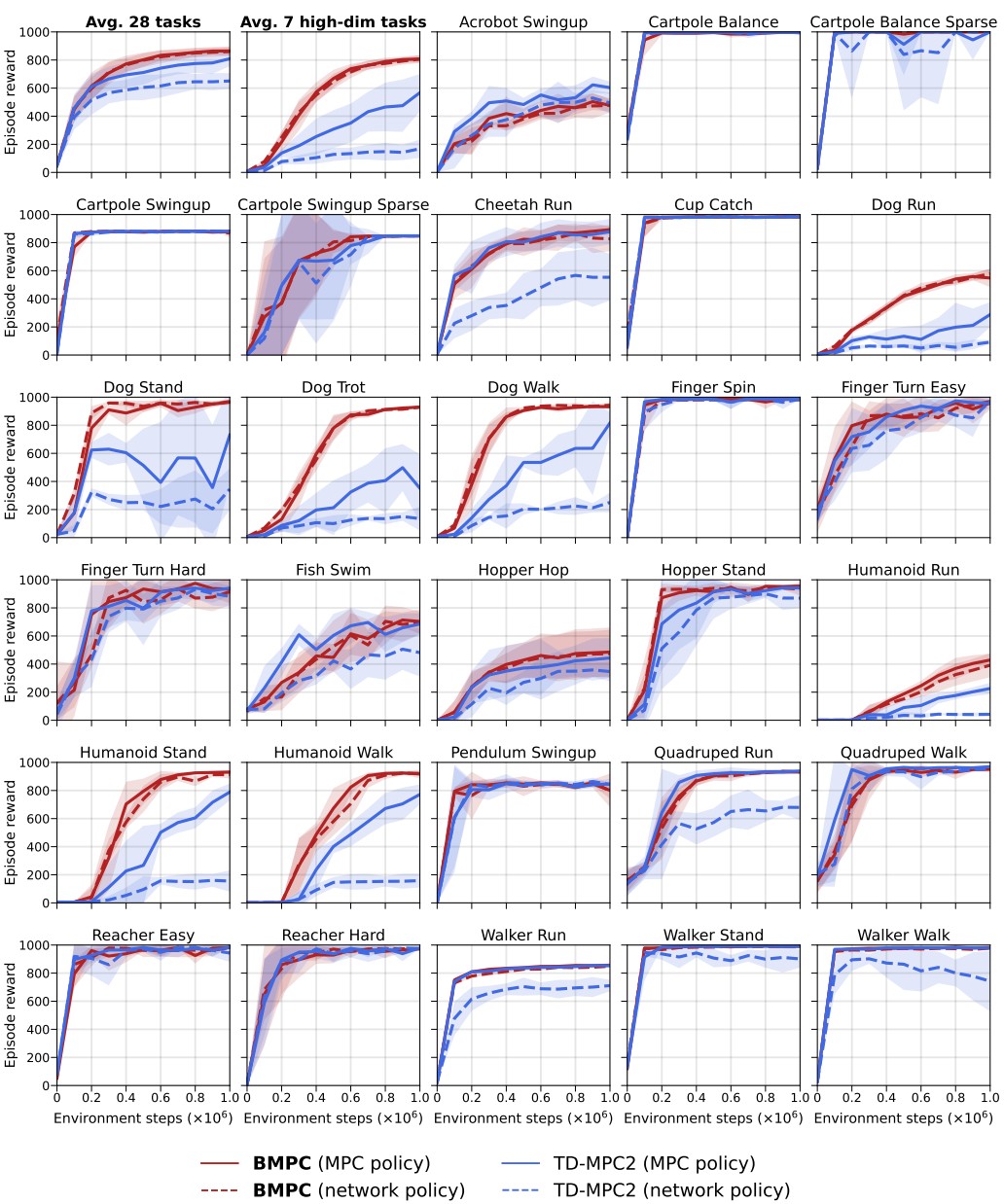

Figure 10: **Performance of different policies on DMControl.** Evaluation performance of the network policy compared to the MPC policy in BMPC and TD-MPC2. In the top left, we present the average performance of 7 high-dimensional locomotion tasks and all 28 tasks. Mean and 95% CIs over 5 seeds.

# D    TASK VISUALIZATION

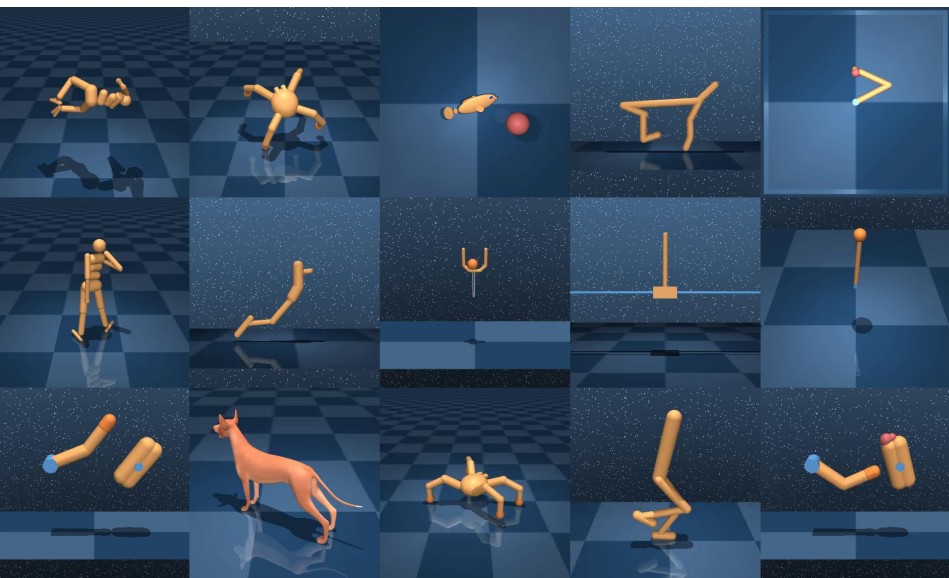

Figure 11: **DMControl tasks visualization.** Images of all the embodiments we control in the DM-Control tasks. The tasks include controlling them to run, walk, jump, balance, reach, and perform actions like swing-up and spin, covering a diverse range of continuous control scenarios.

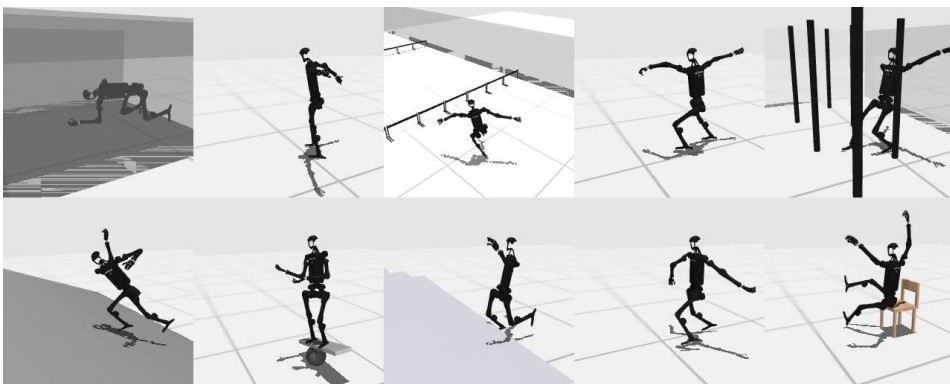

Figure 12: **HumanoidBench locomotion suite visualization.** Images of the Unitree robot we control in the HumanoidBench locomotion suite. The tasks include running, walking, crawling, balancing, sitting, reaching, and performing actions like walking on stairs or walking while avoiding collisions with poles, which cover a diverse range of robotic locomotion scenarios.

