# OpenReview forum: "Bootstrapped Model Predictive Control"
_ICLR.cc/2025/Conference — ICLR 2025 Poster_

### Official Review · Reviewer_Ayiq · 2024-11-01

**Soundness:** 3
**Presentation:** 3
**Contribution:** 2
**Rating:** 6
**Confidence:** 3

**Summary:**

This paper presents BMPC, a plan-based MBRL algorithm that integrates expert imitation for policy learning, performs model-based TD-learning for value learning, and introduces lazy reanalyze to utilize re-planning results better.

**Strengths:**

The paper is well-written and easy to follow.

For high-dimensional locomotion tasks, the paper presents ablation experiments comparing MPC policy with network policy. The results indicate that learning a network policy through expert imitation significantly enhances the performance of MPC.

The approach is evaluated against several baseline agents on DeepMind Control Suite environments.

**Weaknesses:**

1. The paper relies on TD-MPC2 and introduces modifications such as expert imitation and lazy reanalysis. However, these improvements seem overly engineering-focused and do not provide significant insights into the research direction.

2. It is unclear how the expert policy π is initially established as the prior policy. Could you describe how expert policy π evolves during training?

3. In the experimental section, the authors use only 3 seeds as following TD-MPC2. However, I believe that this is not enough. Could you provide additional results with more seeds as DreamerV3 if possible?

4. The related work section provides a rather simplistic overview of MPC research. I recommend enhancing this section for greater comprehensiveness.

**Questions:**

1. The paper replaces the Q-network with a V-network. What are the specific implications of using either Q or V for the training of the algorithm?

2. In line 260, the paper states that "setting N too large would lead to excessive compounding errors," yet it seems that increasing N would be less biased for the V-value estimation. Could you clarify this contradiction?

3. In line 325, could you provide a detailed explanation of "environment step"?

---

> ### Author Response · Authors · 2024-11-18
>
> **1. Overly engineering-focused modifications and limited research insights.**
>
> > These improvements seem overly engineering-focused
>
> We respectfully disagree with the idea that our work focuses too much on engineering. Here’s why:
>
> (1) We did not tune any parameters of TD-MPC2. For the TD-MPC2 components, we used the exact same hyperparameters as in the original work. BMPC's hyperparameters were also not extensively tuned, as their values were determined through theoretical analysis and minimal experimentation based on their well-understood nature.
>
> (2) Our approach does not simply replace modules of TD-MPC2 with "better" ones or add tricks. Instead, we addressed a specific issue—performance gap between policies—by proposing a simple solution inspired by expert iteration. This is not a trivial "A+B" combination, as expert iteration is a conceptual approach aimed at policy improvement. Applying it to data-driven MPC method requires careful design choices.
>
> (3) While our method incorporates techniques such as imitation and replanning, which may seem more "engineering-oriented", we believe that the problem itself can indeed be addressed through such practical solutions. Sometimes, simplicity is a strength, and introducing new theories may not be needed.
>
> > do not provide significant insights into the research direction.
>
> We believe that the issues we identified and the solutions we proposed offer meaningful research insights.
>
> The strong performance of TD-MPC2 motivates us to explore further ways to leverage MPC's capabilities, such as through a bootstrapping approach. Our proposed expert imitation and lazy reanalyze methods are reflections of these research insights. Additionally, the BMPC architecture can be combined with gradient-based policy optimization and can also be applied in offline RL settings, which are further manifestations of our research insights.
>
> Moreover, the issue we identified in TD-MPC2 could be explored further. We hypothesize that it may be related to the comparative strengths and weaknesses of gradient-based versus sample-based optimization in high-dimensional action spaces, as discussed in our response to reviewer tbYo.
>
> While it's easy to make claims, proving them is hard. So, we've avoided overemphasizing unverified insights and instead focused on solving the identified problem with solid experimental results to show that we've addressed it.
>
> **2. Describe how expert policy π evolves during training.**
>
> If we understand correctly, you are asking how the expert policy (the MPC policy), evolves and improves during training.
>
> The MPC policy π is derived through MPPI planning, based on the world model, value function, and network policy. The process is described in the "MPC with a policy prior" part in the Background section of our paper.
>
> The performance of the expert policy improves as both the world model and value function become more accurate through learning. The more accurate the world model and value function, the stronger the expert policy becomes.
>
> While the specific mechanism of how MPPI makes decisions is not the primary focus of our work, the details can be found in the original [TD-MPC paper](https://arxiv.org/abs/2203.04955), where *Algorithm1* explains the MPPI planning process in detail.
>
> **3. Small number of seeds in experiments.**
>
> We followed the TD-MPC2, which used 3 seeds for experiments. From our results, the 95% CI for most curves are fairly narrow, so we believe that 3 seeds are sufficient to support the experimental findings.
>
> However, since two reviewers are concerned about the number of seeds, we will increase the number of seeds to 5 for all experiments. This may take some time, but we will do our best to upload the revised paper with updated results before the discussion period ends (Nov 26).
>
> Additionally, we have already conducted experiments on more domains. Specifically, we extended our experiments on [HumanoidBench](https://arxiv.org/abs/2403.10506) to verify whether BMPC's performance holds in controlling more complex embodiments. This benchmark requires the agent to control a Unitree robot, with a large action space (61-dimensional). All BMPC hyperparameters/settings remain the same as those used on DMControl.
>
> The results show that BMPC maintains superior performance on the HumanoidBench locomotion suite. We have uploaded the results [here](https://github.com/bmpc-anonymous/bmpc), and we will include these new results in the revised paper.
>
> **4. Suggestion to mention related work on MPC research.**
>
> We appreciate the suggestion to include more related work on MPC research. We will incorporate the references suggested by reviewer qEhR to improve the related work section. The revised paper, along with other changes, will be uploaded before the discussion period ends (Nov 26).

---

> ### Author Response · Authors · 2024-11-18
>
> **5. What are the specific implications of using either Q or V for the training of the algorithm?**
>
> We did not simply replace Q with V; rather, we changed the way of value learning, moving from off-policy TD-learning (Q-iteration) to on-policy model-based TD-learning.
>
> We summarize the theoretical facts and experimental results related to value learning as follows, to highlight the specific implications of using Q or V:
>
> **Theoretical facts:**
>
> (1) When using Q-iteration, which follows the Bellman optimal equation, the algorithm must utilize a Q-network for value learning. This is the standard approach in DDPG-style methods.
>
> (2) When using on-policy model-based TD-learning, the algorithm can employ either a V-network or a Q-network for value estimation.
>
> (3) In Q-iteration, it is necessary to use ensemble Q-networks, and select the minimum Q-value for computing TD-target. This is to avoid overestimation bias, which is critical for Q-based methods. However, this increases the number of parameters of the Q-network. Most Q-based methods, such as DDQN, Rainbow DQN, and TD-MPC2, adopt similar technique.
>
> **Experimental results:**
>
> (1) When using on-policy model-based TD-learning, there is no significant difference between using a Q-network or a V-network in terms of algorithm performance.
>
> (2) Compared to Q-iteration, on-policy model-based TD-learning improves algorithm performance. See the ablation section of our paper, where variant3 is compared with BMPC.
>
> (3) On-policy model-based TD-learning allows BMPC to avoid using large ensembles of value networks, reducing the number of network parameters.
>
> **6. Clarification on the trade-off between large N and compounding errors.**
>
> When calculating the TD-target, we use the world model to estimate the value of the current policy online. A longer TD-horizon requires the world model to "imagine" further into the future, which can introduce greater compounding errors from the model.
>
> Based on our early experiments, the world model in TD-MPC2 does not perform well in long-horizon predictions, which is why we opted for a shorter TD-horizon (N=1).
>
> We would like to further clarify this issue with the following points:
>
> (1) The "compounding errors" mentioned in our paper refer to the compounding errors of the world model’s predictions. This is a common term used in the MBRL literature.
>
> (2) It is indeed true that increasing N can reduce bias in V-value estimation, but this is only the case when trained with real data rather than imagined data. In MBRL, model errors often have a more significant impact, which prevents us from setting a long TD-horizon unless we have an accurate model.
>
> (3) In fact, although there is a performance difference between N=1 and N=3 on DMControl, it is not substantial.
>
> **7. Detailed explanation of "environment step".**
>
> The relationship between *environment step*, *inference step*, and *action repeat* is as follows: During an *inference step*, the agent selects an action based on its current policy. *Action repeat* refers to the number of consecutive times the same action is applied in the environment without re-evaluating the policy. Each time the action is applied, it results in an *environment step*, where the environment progresses and returns an updated state and reward. When the *action repeat* is greater than 1, multiple *environment steps* are taken for a single *inference step*, meaning the agent does not update its action choice until after repeating the same action for the specified number of steps.
>
> The term "environment step" is commonly used in the broader RL literature, including in works like TD-MPC2 and DreamerV3. In this work, all DMControl tasks use an action repeat of 2. This means that the number of environment steps is twice the number of inference steps.

---

> > ### Comment · Reviewer_Ayiq · 2024-11-28
> >
> > Thank you for your detailed response to my review. The authors' rebuttal has addressed some of my concerns. I have raised my score.

---

> ### Comment · Area_Chair_qXMF · 2024-11-25
> **Please read rebuttal**
>
> Dear Reviewer Ayiq, Could you please read the authors' rebuttal and give them feedback at your earliest convenience? Thanks. AC

---

### Official Review · Reviewer_qEhR · 2024-11-02

**Soundness:** 4
**Presentation:** 4
**Contribution:** 3
**Rating:** 8
**Confidence:** 4

**Summary:**

Current state-of-the-art in MBRL learn a policy and Q-function in a model-free manner, then couple it with online planning via MPC to improve the policy. However, the authors of this paper find that on complex tasks, there is a substantial gap between the policy and policy+MPC performance. They propose a number of modifications, including an expert iteration-like scheme, for improving the final policy. This involves alternating between improving the policy with MPC online and imitating the updated policy as an expert offline. They find that not only does this improve performance, but it also closes the gap between the network policy and combining it with MPC. With their algorithm, BMPC, the policy learned via expert iteration actually nearly matches the performance of coupling it with online re-planning. Therefore, it is a viable final policy which amortizes the online planning with MPC.

**Strengths:**

- Improving the performance of MBRL algorithms by combining model-based and model-free components is a timely and important problem.
- The paper is well organized and overall clearly written. It does a good job explaining the novelty and results and provides enough information to support its claims.
- The analysis of the performance gap between the model-free policy and bootstraped MPC policy in TD-MPC2 is particularly thorough and very informative, doing a good job to motivate the paper.
- The evaluations are very thorough, considering all 28 environments in the DMControl benchmark tasks and an adequate selection of baselines, such as SAC, DreamerV3, and TD-MPC2. There is also a nice outline of questions to be considered in the evaluation section. And the results do indicate that BMPC can outperform these baselines, suggesting it is a promising method for combining the advantages of model-free and model-based RL.
- The lazy reanalyze method proposed in the paper is a nice solution to recomputing expert actions while still making use of a replay buffer. It's also nice that this reanalyze interval is a free hyperparameter rather than coupled directly to the number of updates to the policy.

**Weaknesses:**

- Some details are missing from the exposition, such as how the MPC policy is computed each batch without having to re-run the planning algorithm. One possibility is that the mean and covariance of the Gaussian is stored in the replay buffer. However, the paper would benefit from making this more concrete.
- It would be worth mentioning other work which attempt to imitate an expert MPC controller [1-4]. Alternatively, there are also approaches which attempt to learn residual on MPC [5-6], which is another way of bootstrapping MPC. And there are approaches which attempt to improve MPC via learned sampling distributions [7, 8]. Although not expert iteration, they are definitely related approaches which are important to mention.

References:
[1] Pan et al., "Agile Autonomous Driving using End-to-End Deep Imitation Learning," RSS 2018.
[2] Pan et al., "Imitation learning for agile autonomous driving," IJRR 2020.
[3] Sacks et al., "Learning to optimize in model predictive control," ICRA 2022.
[4] Fishman et al., "Motion policy networks," CoRL 2023.
[5] Sacks et al., "Deep Model Predictive Optimization," ICRA 2024.
[6] Silver et al., "Residual Policy Learning," arXiv 2018.
[7] Power et al., "Variational Inference MPC using Normalizing Flows and Out-of-Distribution Projection," RSS 2022.
[8] Sacks et al., "Learning Sampling Distributions for Model Predictive Control,"  CoRL 2022.

**Questions:**

- How is the MPC policy computed for imitation learning without rerunning the planning algorithm every batch? Is the mean and covariance of the MPC policy stored in the replay buffer as well?

---

> ### Author Response · Authors · 2024-11-18
>
> We thank the reviewer for their positive feedback and for recognizing the strengths of our work. Below, we provide our responses to each of the concerns:
>
> **1. Clarification on how the MPC policy is computed without re-running the planning algorithm.**
>
> Thank you for pointing this out, and you are correct. In our implementation, we store the action distribution of the MPC policy in the replay buffer, specifically the mean and variance of the Gaussian distribution, rather than the actions themselves. We will make sure to clarify this detail in the revised paper.
>
> **2. Suggestion to mention related work on MPC imitation.**
>
> We will gladly include and cite the related work you mentioned, as it is indeed highly relevant to our study! Due to the limitations of our perspective, we hadn’t fully summarized all relevant work, and we truly appreciate your valuable input, which will help improve the completeness of our paper. We will incorporate these changes along with other revisions and upload the updated paper before the discussion period ends (Nov 26).

---

> ### Comment · Area_Chair_qXMF · 2024-11-25
> **Please read rebuttal**
>
> Dear Reviewer qEhR, Could you please read the authors' rebuttal and give them feedback at your earliest convenience? Thanks. AC

---

### Official Review · Reviewer_tbYo · 2024-11-03

**Soundness:** 3
**Presentation:** 3
**Contribution:** 4
**Rating:** 8
**Confidence:** 5

**Summary:**

This paper proposes Boostrapped Model Predictive Control (BMPC), a model-based RL method that improves upon TD-MPC. TD-MPC consists of two policies: a network policy trained with RL, and an MPC policy guided by the network policy and its value function. The authors found the network policy in TD-MPC to perform much worse than the MPC policy. This is attributed to the network policy being trained in a model-free manner, which is less sample-efficient than MPC. The poor network policy then seeps into the value function, which in turn impacts the performance of the MPC policy. To mitigate this issue, the authors propose to train the network policy by directly imitating the MPC policy, while still using the network policy and its value function to guide the MPC policy. This effectively "distills" the MPC policy into a neural network, avoiding the policy mismatch in TD-MPC. However, this kind of imitation introduces significant overhead into policy learning, as each training step requires replanning on all samples. To address this, the authors propose Lazy Reanalyze, which replans on a small set of transitions every few training steps. With Expert Imitation and Lazy Reanaylze, BMPC achieves better asymptotic performance in fewer number of environment steps compared to TD-MPC on high-dimension DM Control tasks, all without introducing significant overhead.

**Strengths:**

1. The paper is motivated by a thorough analysis of the TD-MPC algorithm, which crystalizes into a key insight: the performance of TD-MPC is impacted by a mismatch between the network policy and the MPC policy. The proposed algorithm then follows as an intuitive solution. The overall flow of the paper is excellent, making it a pleasant read.
2. The empirical results of BMPC demonstrate a significant performance gain over TD-MPC, especially in complex high-dimensional locomotion tasks. In most tasks, BMPC is more sample-efficient than TD-MPC. And in a few tasks, it also achieves higher asymptotic performance.
3. The ablation experiments supplement the main experiment well, providing insights into the effect of hyperparameters and key design decisions. This adds to the credibility of the method
4. Overall, I believe this paper proposes a simple method that brings a new perspective to model-based RL. Therefore, I recommend acceptance.

**Weaknesses:**

1. The expert imitation procedure introduces overhead into the training pipeline, as each training step requires replanning. Although this is sidestepped by Lazy Reanalyze, it remains a fundamental limitation of the method.
2. The experiments are run with a small number of seeds (3 seeds).
3. The experiments succinctly prove the point that the authors try to make. That said, it would strengthen the paper to include experiments across more diverse domains (those in TD-MPC 2).

**Questions:**

1. Are there other reasons behind TD-MPC's bad value function? If it's just the sample inefficiency model-free policy optimization, then the network policy should eventually catch up to the MPC policy. But this is not the case as shown in Figure 2 -- the asymptotic performance of the network policy is worse.
2. What is the tradeoff of pure MPC policy distillation compared to model-free policy optimization? Are there domains or settings where BMPC performs worse than TD-MPC? If so, what are the reasons?
3. Can you describe the three variants in Figure 6 (a) in more detail? I would also suggest adding the details of the variants in the appendix.

---

> ### Author Response · Authors · 2024-11-18
>
> **1. Expert imitation introduces overhead into the training pipeline.**
>
> We agree that this is true, and unfortunately, unavoidable. Although we have addressed this issue by lazy reanalyze, which has proven to be remarkably effective. Without lazy reanalyze, the training time for BMPC would be 10 times longer.
>
> There are also potential ways to further improve this, such as further parallelizing the reanalyze process by adopting a distributed architecture similar to [EfficientZero](https://arxiv.org/abs/2111.00210). While this approach does not reduce the compute overhead, it can decrease the overall training time by scaling up the computation, indicating that BMPC is scalable and can be easily parallelized.
>
> In fact, the re-planning shouldn't introduce this much overhead. The main issue is that MPPI itself is costly. Specifically, for a single inference, MCTS requires 800 simulations in board games in [MuZero](https://arxiv.org/abs/1911.08265), gumbel search only requires 32 simulations in various domains in [EfficientZeroV2](https://arxiv.org/abs/2403.00564), while MPPI requires 512 * 6 * 3 = 9216 simulations. We tried reducing the number of simulations, but this inevitably led to a decrease in performance to some extent.
>
> However, from another perspective, although MPPI incurs a high computational cost, this might be the reason why it performs so well in continuous tasks.
>
> **2. Small number of seeds in experiments.**
>
> We followed the TD-MPC2, which used 3 seeds for experiments. From our results, the 95% CI for most curves are fairly narrow, so we believe that 3 seeds are sufficient to support the experimental findings.
>
> However, since two reviewers are concerned about the number of seeds, we will increase the number of seeds to 5 for all experiments. This may take some time, but we will do our best to upload the revised paper with updated results before the discussion period ends (Nov 26).
>
> **3. Experiments on more domains.**
>
> We have already conducted experiments on more domains. Specifically, we evaluate BMPC on [HumanoidBench](https://arxiv.org/abs/2403.10506) to verify whether BMPC's performance holds in controlling more complex embodiments. This benchmark requires the agent to control a Unitree robot, with a large action space (61-dim). All BMPC hyperparameters remain the same as those used on DMControl.
>
> The results show that BMPC maintains superior performance on the HumanoidBench locomotion suite. Although the HumanoidBench already includes results for TD-MPC2, we re-ran the experiments using the latest code (which yielded better results). We have uploaded the results [**here**](https://github.com/bmpc-anonymous/bmpc), and we will include these new results in the revised paper before the discussion period ends (Nov 26).
>
> **3. Are there other reasons behind TD-MPC's poor value function?**
>
> This is a crucial question! On the surface, we tend to think of it this way:
>
> (1) TD-MPC's network policy is trained similarly to SAC, with only two differences: The training data comes from the MPC policy, which is of higher quality; The inputs are well-represented latent vectors rather than raw observations.
>
> (2) SAC tends to fail on high-dimensional tasks.
>
> (3) Hypothesis: Even with better data and better representations, SAC still fails.
>
> (4) Therefore, TD-MPC's network policy also fails.
>
> As for why SAC fails, one potential reason is that for high action-dim tasks, sample-based optimization (like MPPI) is more robust and suitable than gradient-based optimization (like DDPG-style max-Q gradient).
>
> In fact, we tried to combine both expert imitation and max-Q gradient for policy optimization, and it showed fairly good results. However, we have not yet explored this approach in depth. We believe there are many more things to think about, and more experiments are needed to uncover the underlying reasons behind the phenomenon.
>
> **4. What is the tradeoff of pure MPC policy distillation compared to model-free policy optimization?**
>
> Despite attempts to find any drawbacks, we haven’t identified any clear disadvantages or tradeoffs of MPC policy distillation compared to gradient-based policy optimization in our experiments.
>
> Besides DMControl and HumanoidBench, we also tested a few tasks from MyoSuite and ManiSkill. Overall, BMPC’s performance didn’t fall behind TD-MPC2 in any of these domains. In certain tasks, BMPC might fall behind a little due to an “inappropriate” network policy std, but we found that this can be fixed by adjusting the task-specific entropy loss coefficient (which we did not do), so it’s not a real tradeoff.
>
> In tasks where the MPC policy is weaker than a SAC-like policy (possibly due to a poorly learned world model or an inappropriate planning horizon), BMPC will probably perform worse than TD-MPC2. That said, we have yet to encounter such a situation.

---

> ### Author Response · Authors · 2024-11-18
>
> **5. Describe the three variants in Figure 6(a) in more detail.**
>
> **Facts:**
>
> A network policy can be learned by:
>
> (1a) Max-Q gradient; (1b) Expert imitation;
>
> A network policy is needed for:
>
> (2a) Computing TD-target; (2b) Guiding the MPPI process;
>
> The value function can be learned by:
>
> (3a) Off-policy TD-learning (Q-iteration); (3b) On-policy model-based TD-learning;
>
> **Variants:**
>
> Variant 1: Learn network policy A through (1a), learn network policy B through (1b); use A for (2a), use B for (2b); learn the value function through (3a).
>
> Variant 2: Learn network policy A through (1a), learn network policy B through (1b); use A for (2a), use both A and B for (2b); learn the value function through (3a).
>
> Variant 3: Learn network policy A through (1b); use A for both (2a) and (2b); learn the value function through (3a).
>
> BMPC: Learn network policy A through (1b); use A for both (2a) and (2b); learn the value function through (3b).
>
> We will add further details of these variants in the appendix. Similarly, we will upload the revised paper with these updates before the discussion period ends (Nov 26).

---

> > ### Comment · Reviewer_tbYo · 2024-11-25
> >
> > Thanks for addressing my comments! I maintain my evaluation of the paper.

---

### Official Review · Reviewer_4Ngs · 2024-11-05

**Soundness:** 3
**Presentation:** 3
**Contribution:** 3
**Rating:** 6
**Confidence:** 3

**Summary:**

The paper proposes Bootstrapped Model Predictive Control (BMPC), a model-based RL method that combines MPC with a policy, where the policy gives MPC better prior distribution of action to sample over, while the MPC uses a refined plan to train this policy. The authors propose lazy reanalyze method to make imitation of MPC policy more efficient, by eliminating the frequency of replanning & immediate update. The authors study the performance of proposed BMPC on a variety of continuous control tasks, compared against SOTA MBRL methods as well as model-free methods.

**Strengths:**

The paper is clearly written and the method is well-motivated: the policy can make MPC search more efficient while MPC's high-quality results can help with policy learning.

The result and analysis are solid and convincing. The authors benchmarked on a good number of environments & standard benchmarks.

The good ablations answer a lot of my questions, and I expect all readers to find them very informative.

**Weaknesses:**

The outstanding concern is the legitimacy of letting MPC & policy bootstrapping each other. For example, it's unclear what influence this has on exploration. Will the added policy hurt the exploration of MPC? Could you highlight how exploration is achieved in your model or is it just like in TD-MPC2?

The authors chose Dreamer v3 as the MBRL baseline in the main evaluations. Since Dreamer v3 is specifically designed for visual observation & multi-tasking, I wonder whether the authors tried tuned it for state-based evaluations.

**Questions:**

In terms of the trade-off between exploration & exploitation, do you have to carefully tune/have a curriculum on the sigma std for the MPC?

What's the effect of EMA in Eqn(7)? Is it mainly for stabilization? How grounded is this design choice?

---

> ### Author Response · Authors · 2024-11-18
>
> **1. Concern on bootstrapping between MPC and policy affecting exploration/exploitation.**
>
> > Will the added policy hurt the exploration of MPC?
>
> No, overall it does not.
>
> First, we observed that adding the guiding policy improves the performance of MPC compared to not adding it, similar to the findings in the TD-MPC2.
>
> Second, entropy loss in policy training prevented potential harm to MPC exploration. In our early experiments, we found that if the policy std is too small, the guiding trajectories may become too narrow, leading to premature convergence to local optima. However, This issue can be mitigated by simply adding entropy loss to policy learning, as in BMPC.
>
> For clarity, the main reason we add exploration in lazy reanalyze was to prevent the bootstrapping mechanism from causing the network policy’s std to shrink over time. In most tasks, this has no impact on performance. To be specific, we aimed to avoid the following scenario:
>
> (1) The std of the MPC policy is always smaller than that of the network policy, since the samples generated by the network policy are iterated by MPPI multiple times.
>
> (2) The network policy mimics the MPC policy, including its std, causing the network policy’s std to shrink.
>
> (3) Eventually, the network policy’s std converges to the minimum value allowed by the MPC policy, which may not be desirable.
>
> Thus, we enforced a lower bound on the std of the network policy during lazy reanalyze to prevent this from happening. However, this situation rarely occurs in most tasks, and even when it does, it typically doesn’t affect performance negatively. It only impacts hard-exploration tasks with sparse rewards a little.
>
> > Could you highlight how exploration is achieved in your model, or is it similar to TD-MPC2?
>
> We use the same MPPI configuration as TD-MPC2, including the exploration part. The MPC exploration setting in TD-MPC2 is quite simple but generalizable, which is why we didn’t make any modifications to it.
>
> > Do you have to carefully tune or use a curriculum for the std (sigma) of the MPC?
>
> No, we didn’t need to tune anything. We used the same MPPI hyperparameters and configuration as TD-MPC2, including the std. In fact, we kept all TD-MPC2-related hyperparameters in BMPC unchanged to ensure a fair comparison.
>
> **2. Concern on the appropriateness of using DreamerV3 for state-based comparisons.**
>
> It is true that DreamerV3 was primarily designed for visual tasks (to my knowledge, it doesn’t support multi-tasking). Nevertheless, it also reports state-based DMControl results in the [paper](https://arxiv.org/abs/2301.04104), which is the same as our setting. Furthermore, the [official code](https://github.com/danijar/dreamerv3) provides hyperparameters for state-based DMControl, so we did not conduct any additional tuning.
>
> **3. What's the Effect of EMA in Eqn (7)?**
>
> The main purpose of the EMA in Eqn (7) is to normalize the KL loss used for imitation, thereby balancing the KL loss with the entropy loss. Depending on the task, the value of the KL loss of a Gaussian can vary widely, from 0 to 1000, which introduces instability during training and makes it difficult to apply a fixed entropy loss coefficient across different tasks.
>
> > How grounded is this design choice?
>
> The similar design choice has been used in both DreamerV3 and TD-MPC2, where it serves the same purpose of balancing the actor loss and entropy loss. Therefore, it is a well-validated design choice based on extensive experimentation.

---

> ### Author Response · Authors · 2024-11-19
>
> By the way, we also conducted new experiments on [HumanoidBench](https://arxiv.org/abs/2403.10506), which has a larger action space(61-dim), and BMPC's performance remained strong without any changes to the hyperparameters. This might help illustrate that the BMPC approach does not excessively interfere with MPC's exploration. We have uploaded the results [here](https://github.com/bmpc-anonymous/bmpc), and we will include these new results in the revised paper before the discussion period ends (Nov 26).

---

> ### Comment · Area_Chair_qXMF · 2024-11-25
> **Please read rebuttal**
>
> Dear Reviewer 4Ngs, Could you please read the authors' rebuttal and give them feedback at your earliest convenience? Thanks. AC

---

### Author Response · Authors · 2024-11-24
**Revised Submission: Expanded Experiments, Increased Seeds, and Additional Clarifications**

We have uploaded a revised version of the paper, incorporating both additional experiments and revisions based on the reviewers' valuable feedback. Below, we summarize the key changes in the updated manuscript:

**1. New Experiments on HumanoidBench**

We conducted additional experiments to evaluate BMPC on HumanoidBench, a benchmark requiring control of a Unitree robot with a high-dimensional action space (61 dims). All BMPC hyperparameters remain the same with those used on DMControl.

- The results demonstrate that BMPC maintains superior performance on the HumanoidBench locomotion suite.

- While HumanoidBench already provides results for TD-MPC2, we re-ran their experiments using the latest code (which yielded improved results) for a fair comparison.

**2. Increased Evaluation Seeds**

As suggested by reviewers tbYo and Ayiq, we increased the number of seeds for all experiments from 3 seeds to 5 seeds to ensure greater statistical robustness.

**3. Details on BMPC Variants**

As suggested by reviewer tbYo, we added detailed descriptions of BMPC variants in the appendix to provide more comprehensive insights.

**4. Expanded Related Work**

As suggested by reviewers qEhR and Ayiq, we expanded the Related Work section to include a discussion of methods related to imitating and enhancing MPC and added relevant citations.

**5. Details on Imitation Target**

In response to reviewer qEhR, we clarified the implementation details of the imitation target in the manuscript.

We hope these revisions address the reviewers' concerns and further improve the quality of the paper. We welcome any additional feedback.

---

### Meta-Review · Area_Chair_qXMF · 2024-12-17

**Metareview:**

This paper proposes to bootstrap imitation learning policies with MPC motions. The MPC gives a target for neural policy while the neural policy gives back a good prior for the MPC. The proposed approach achieves SOTA performance on various challenging benchmarks. The idea is novel. The experiments are solid. The contribution is genuine. The overall quality is above the bar of ICLR.

**Additional Comments On Reviewer Discussion:**

The authors addressed most of the concerns.

---

### Decision · Program_Chairs · 2025-01-22

Accept (Poster)